# Foodborne Diseases: A Study before and during the COVID-19 Pandemic in Brazil

**DOI:** 10.3390/nu16010060

**Published:** 2023-12-25

**Authors:** Fernanda Vinhal Nepomuceno, Rita de Cassia Coelho de Almeida Akutsu, Cainara Lins Draeger, Izabel Cristina Rodrigues da Silva

**Affiliations:** 1Department of Nutrition, Faculty of Health Sciences, University of Brasilia, Brasilia 70910-900, Brazil; fernanda.vinhal@hotmail.com (F.V.N.); rita.akutsu@gmail.com (R.d.C.C.d.A.A.); 2Faculty of Ceilândia, University of Brasilia, Brasilia 72220-275, Brazil; belbiomedica@gmail.com

**Keywords:** national survey, foodborne disease, public health, health survey

## Abstract

Foodborne Diseases (FBDs) are a worldwide problem and occur after contaminated food has been ingested, signaling a lack of food quality. Even though the SARS-CoV-2 virus is not transmitted through food, the COVID-19 pandemic has caused several challenges worldwide that have had direct implications on food production and handling, stimulating and reinforcing the adoption of good manufacturing and food handling practices. The aim of this study was to analyze data on notifications of FBD in Brazil in the years before (2018 and 2019) and during (2020 and 2021) the COVID-19 pandemic. Secondary data from the National System of Notifiable Diseases was analyzed, evaluating: overall incidence rate, lethality and mortality, contamination sites, and criteria for confirming the etiological agent. There were 2206 records of FBDs, and the mortality rate was 0.5% in both periods. The incidence rate before the pandemic was 6.48 and during the pandemic was 3.92, while the mortality coefficient was 0.033 before and 0.019 during the pandemic, both per 100,000 inhabitants. There was no significant difference in the number of FBD notifications in the evaluated periods. There was a migration of the location of FBD, with a significant increase in FBD notifications in hospitals and health units and a reduction in notifications from social events. There was a significant increase in the type of criteria used to confirm outbreaks, with an increase in clinical laboratory tests and clinical reports for bromatology. The increase in notifications in hospitals and health units demonstrates the necessity of improving food safety knowledge and the attitudes and practices of food handlers and healthcare professionals.

## 1. Introduction

COVID-19 is caused by a new coronavirus pathogen. The infection causes an acute febrile respiratory illness. People with COVID-19 can be asymptomatic and silently spread the disease [1]. Contact with a contaminated air environment is the main factor in the spread of the disease. Among the various potential atypical modes of transmission, food-borne transmission is widely discussed [2]. Presently, there is no confirmation that COVID-19 is a foodborne disease. However, food contamination is a serious health and management problem. Since hand contamination can result in the transport of pathogens to the oral or nasal cavity, it has been widely reported that good hand hygiene during the COVID-19 outbreak, or not, was, and is, essential to prevent cross-contamination. An infected person can contaminate the entire surrounding environment [3].

The public health challenge was not only to mitigate the spread of the virus but also to guarantee a safe, continuous, and high-quality food supply. Despite the socioeconomic adversities, it was essential that the food chain, from cultivation to consumption, was meticulously monitored, emphasizing employee health, hand sanitization, surface and environmental sanitation, and respect for social distancing [4,5].

According to the Food and Agriculture Organization (FAO), there is no evidence of the transmission of the COVID-19 virus through the food chain [6]. It is necessary to apply the principles of environmental sanitation, personal hygiene, and food safety practices to achieve prevention. The integrity of the food process depends on strict observance of good hygiene practices, environmental sanitation, and proper procedures for handling food, especially of animal origin [5].

In Brazil, the National Health Surveillance Agency (ANVISA) expanded its guidelines for food companies through Technical Notes (NT) No. 47, 48, and 49/2020, emphasizing that, although there is no proof of transmission of the SARS-CoV-2 virus through food products, there was a need for greater attention to be paid to good manufacturing and product handling practices, with a focus on workers and handling environments [7,8]. The recommendations call for the use of masks and gloves in food services, as well as an assessment of workers’ health, personal hygiene, use of physical barriers, personal protective equipment, physical distancing, control of raw materials, controlled production flow, division of shifts for employees, and transportation of products [7,8].

Global public health bodies have always been on high alert regarding foodborne diseases (FBDs), given their transmissibility and the complexity of screening for them [9,10,11]. According to a 2015 WHO report [12], around 600 million cases of FBDs and 420,000 deaths are reported worldwide every year due to bacteria, viruses, parasites, poisons, and chemicals. In Brazil, the Health Surveillance Secretariat recorded 7630 outbreaks of foodborne diseases between 2007 and 2017, totaling 134,046 people sickened by foodborne diseases, 19,394 hospitalizations, and 127 deaths [10].

Despite growing international awareness of FBDs as a major risk to the health of the population and the socioeconomic development of the country, food safety continues to be marginalized [10]. One of the main obstacles is the need for more accurate data on the extent and cost of foodborne diseases in the country. This data would allow public officials to define priorities for public health actions in terms of preventing and treating these diseases. Epidemiological data on foodborne diseases is still scarce, particularly in developing countries. Even the most visible foodborne outbreaks often go unnoticed because they are not reported or adequately investigated [11,12].

It was only in 2007 that Brazil’s Health Surveillance Secretariat developed the National Epidemiological Surveillance System for Foodborne Diseases (VE-DTA), aiming to reduce the incidence of FBDs in the country, based on knowledge and identification of the magnitude of the problem [13].

In Brazil, the current National Epidemiological Surveillance System calls for the notification of cases of notifiable diseases and outbreaks of any etiology. FBD surveillance moves towards collecting information about and investigating outbreaks. However, little is known about the epidemiological profile of foodborne illnesses. Only a few Brazilian states and/or municipalities have statistics and data on the most common etiological agents, the foods most frequently involved, the population most at risk, and the factors contributing to illness [13,14].

This is the first study with a Brazilian national base to analyze data on FBD notifications, comparing the pre-pandemic periods (2018 and 2019) and during the COVID-19 health emergency (2020 and 2021), and considering the implications of the COVID-19 pandemic scenario for food safety in Brazil, the consequent reorganization of health services, food production, and consumption. Some other countries have analyzed the pre-pandemic scenario and the scenario during the pandemic, but they did not analyze the time period of two years before and two years during the pandemic.

## 2. Materials and Methods

This research is characterized as a retrospective cohort study and was carried out via the analysis of secondary data from the National System of Diseases and Notifications (SINAN—NET), available from the Brazilian Ministry of Health. The system has a platform where notifications of foodborne illnesses in Brazil are compulsorily recorded, and regulated by specific legislation. This information is publicly available, and therefore there is no need for an agreement or consent form [10,13,14].

The use of secondary data from Health Information System has the advantages of broad population coverage, low cost of collecting information, and ease of longitudinal follow-up [15].

We analyzed all the data entered the SINAN-NET platform [8,11,12] of people affected by an FBD between 2018 and 2021 at two points: two years before (2018 and 2019) and two years during the declaration of the COVID-19 pandemic (2020 and 2021).

To systematize the information, the Brazilian Manual for the Prevention and Control of FBDs lists five confirmation criteria for investigating the etiological agent causing the outbreak: (1) clinical laboratory criterion, when the cause of the outbreak is concluded from the results of clinical samples; (2) clinical epidemiological, which occurs in the absence of clinical and bromatological samples taken, in negative laboratory results, or laboratory results incompatible with the clinical presentation and epidemiology of the outbreak; (3) bromatological laboratory, in which there is the result of the bromatological sample; (4) clinical bromatological laboratory in which the outbreak is closed with the identification of the same etiological agent in both the clinical and bromatological samples; and finally, the (5) inconclusive criterion in which there is no information that allows the outbreak to be closed [13]. 

The same protocol was used as in the study developed by Draeger et al. [12] using Brazilian legislation as a reference. The variables are part of a standardized form with (i) mandatory items, in which the absence of these items does not allow the notification to be recorded in the system’s database; (ii) essential items, which present important data for investigating VE-DTA and calculating epidemiological indicators, but are optional; and (iii) complementary items, which are included in the system to help understand VE-DTA, but are neither mandatory nor essential. These three items can complement the information about each individual case [14].

Additionally, the Integrated Manual for the Prevention and Control of Foodborne Diseases is a document used to regulate the actions and instruments used in the investigation of FBD outbreaks, as well as guiding the information flow of the VE-TDA System, serving as a technical basis for the development of activities. This document defines those exposed as the group of people who participate in a meal which has caused an FBD outbreak (page 74), and sick people as the individuals who have symptoms and are related to the outbreak (pages 74 and 84). The report also informs us that it is usual for an outbreak to include people who consume the food and do not become ill, which may be caused by the resistance and susceptibility of the host, or by consuming portions with non-infectious doses, among others [13]. The characteristics of FBDs were identified, such as the overall incidence rate, the lethality rate, and the mortality rate, contamination sites, and confirmation criteria in investigating the etiological agent causing the outbreak. The lethality rate of FBDs is calculated as the ratio between the number of deaths from FBD and the total number of cases in the period. The FBD incidence rate and the FBD mortality rate were calculated considering the number of registered cases and the national average population during the period investigated, presenting the results for every 100,000 inhabitants [16]. Statistical analysis occurred using the SPSS^®^ program (version 26.0). The Chi-square test was used to assess the associations between categorical variables, with a 95% confidence interval [17].

## 3. Results

This study provides the first overview of Brazilian national FBD data before and during the declaration of the COVID-19 pandemic. It analyzes the information entered into the SINAN-NET platform for everyone affected between 2018 and 2021. 

The case fatality rate was 0.5% in both groups. The incidence rate was 6.48 (per 100,000 inhabitants) before the pandemic and 3.92 (per 100,000 inhabitants) during the pandemic. The mortality coefficient was 0.033 (per 100,000 inhabitants) before and 0.019 (per 100,000 inhabitants) during the pandemic.

It is important to note that, during this period (2018–2021), there were 2206 cases of FBDs registered with the VE-DTA. Of the notifications registered before the pandemic, the average number of people exposed was 55; of these, 16 fell ill. While during the pandemic, the average exposure was 33, and of these, 17 fell ill (Table 1 and Table 2).

The median number of sick people before and during the COVID-19 pandemic was six and seven, respectively, with no recorded deaths. In addition, the only difference shown was between those exposed in the two periods. However, this difference was not significant (Figure 1).

Analyzing the events of FBD contaminations, there was no significant difference in the number of notifications before and during the pandemic. However, there was a significant difference in where the outbreak occurred, with a reduction in cases at social events (*p* = 0.014) and an increase in notifications at hospitals and health units (*p* = 0.031) when the two periods were compared, as seen in Figure 2.

Figure 3 shows a significant difference between the two periods (before and during the pandemic) for the type of test/criteria used to confirm FBD outbreaks. There was a significant increase in the use of clinical laboratory tests (*p* = 0.005) and bromatological clinical laboratory tests (*p* = 0.001), as well as a significant reduction in ignored, inconclusive, or unrecorded cases (*p* = 0.025).

## 4. Discussion

The declaration of the COVID-19 pandemic has generated global concern about food safety, physical and mental health, and obviously, about a SARS-CoV-2 virus about which we knew little about in terms of how it spreads and is treated. The efforts made by Brazil and the rest of the world to adapt hygiene measures and prevent the spread of the COVID-19 virus, with a focus on providing citizens with safe, permanent, and high-quality food, have shown the fragility of recording cases of FBDs, even when it is compulsory. The low level of notification and dissemination of FBD outbreaks pointed out by Gibbons et al., 2014 is one of the significant challenges in making decisions about the measures adopted as public health measures [18].

In the case of Brazil, despite the social and economic difficulties facing the population in general and the country in particular, safety measures were proposed for the food sector, with more extraordinary precautions for each stage of the supply chain, from field to plate. Among the measures adopted were worker health conditions, the use of gloves, masks, and caps to cover hair, which have already been in place since 2004 in compliance with Brazilian legislation [19]. In addition, surface disinfection was carried out more frequently and with stricter control during the delivery of food and meals, especially in the delivery and takeaway system, which changed the hygiene standards of packaging and packaging supports and social distancing [4,5,6,9,20]. 

Regarding the mortality coefficient of the periods evaluated in this study, it was found that both were lower than 0.06 per 100,000 inhabitants, which is lower than that previously found in Brazil in the periods 2007–2017 [10]. In the European Union, there was also a reduction in the number of reported deaths when comparing 2020 with the period 2017–2019, from 30.1% (average of 44 deaths per year in 2017–2019) to 8.8% (34 deaths in 2020) [21]. Given the historical reduction in notifications and the number of deaths, several studies have questioned whether there was, in fact, a reduction in FBDs or a deficiency in notification during the COVID-19 pandemic. Further comparative studies are needed as health systems return to pre-pandemic conditions, as we hope that the markers will evolve positively [21,22,23].

In the two periods evaluated in this study, the lethality rate remained at 0.5% per 100,000 inhabitants, demonstrating that the advent of the COVID-19 pandemic had no impact on the risk of death from FBDs. The rate found in the study by Draeger et al., 2018 is lower, at 0.09% per 100,000 inhabitants over the 11 years investigated [10]. This can probably be explained by the overload of health services with COVID-19, which reached more than 37 million cases, more than 700,000 deaths, and a lethality rate of 1.9. These data shed light on what the priority was during the period in question [24].

Although the expectation, considering the social isolation measures adopted in Brazil, which closed schools, restaurants, and other establishments considered non-essential, was that the results of this study would show a significant difference in notifications and cases, there was only a change in the place of occurrence of outbreaks for hospitals and health services and for events (positive and negative, respectively). The expectation is that due to the lockdown, most occurrences during the pandemic would be domestic cases, which can be found in this study.

It is important to note that even though there were no significant changes in the occurrence of domestic outbreaks between the two periods, these outbreaks represented 40.2% and 37.9%, before and during, respectively, which shows the importance of campaigns, among other measures, to reduce domestic events.

The need for non-pharmacological measures to prevent infection with the SARS-CoV-2 virus has been widely publicized. Measures such as social distancing, wearing masks, cleaning and disinfecting environments, and, especially, hand hygiene, used to combat the SARS-CoV-2 virus, would also be responsible for interrupting the transmission of other viruses and bacteria, such as FBDs, which would lead to a paradigm shift in the levels of transmission and records of FBDs [25].

In addition to the underreporting pointed out by Gibbons et al., 2014, in their study Draeger et al., 2018 already pointed out that there was also, in the Brazilian case, the issue of incomplete data caused by the difficulty in using computerized data consolidation systems and the lack of homogeneity in filling these out by health services in the various administrative instances [14].

In the same vein, using data from an interrupted time series, Nash et al., 2022 found that, in England, the non-pharmaceutical interventions imposed during the COVID-19 pandemic significantly reduced the incidence of endemic diseases such as measles, whooping cough, mumps, and meningitis, as well as incidents of food poisoning [26]. The most significant reduction in England was seen in weekly measles cases, with a percentage reduction of 90.5% (95% CI; 86.8–93.1) from forty-two to five cases per week, followed by cases of whooping cough (90.1% reduction), mumps (88.2% reduction) and meningitis (82.8%). It is valid to mention that the reduction in the total number of food poisoning incidents was 56.4% (95% CI; 42.5–54.2), from 191 cases per week to 83 cases, the lowest percentage reduction.

To justify this difference in the percentage of food poisonings compared to other diseases, the authors point out that food services remained open, even with home delivery or takeaway, offering a potential route of transmission for FBDs [26]. As already highlighted, Draeger et al., 2018 point out that most FBD cases are of domestic origin, which demonstrates a severe lack of health education and adequate knowledge of food preparation and storage in the Brazilian population in general [10].

A study carried out in the United States in 2021 showed awareness on the part of the population that other people touching their food and food products can be a means of transmitting pathogens and possible infections. On the other hand, incorrect information has spread, such as the need to sanitize with soap. This highlights the importance of involving the population in education and communication about food safety [27].

In the European Union, in 2021, a considerable increase was also observed in the proportion of foodborne outbreaks and outbreaks in places of medical care, nursing homes, prisons, and boarding schools, which represented 5.1% during the period 2017–2019, increasing to 7.7% in 2020 and 9% in 2021 [21]. 

Most of the studies presented during this discussion always come to the same conclusion. They always agree that investments are needed in infrastructure (sanitation, water, etc.), in health surveillance of restaurants, hospitals and health services, and in educating the population [21,26,27].

As for the criteria for confirming FBDs, there has been an increase in the investigation of outbreaks based on clinical laboratory and clinical laboratory bromatological criteria, which was to be expected. As the symptoms of COVID-19 were not yet well known, symptoms of FBDs could be confused or overestimated, which required more accurate data using clinical or bromatological samples.

Unlike Brazil, in the United States in 2020, the proportion of clinical laboratory tests was maintained, even with a historic 26% reduction in the incidence of FBDs. This suggests that the change in confirmation criteria has not contributed to reducing the incidence of infection [22]. 

Canada has not yet formally assessed the impact of the COVID-19 pandemic on FBDs, but Dougherty et al., 2023 [28] described that, from March to December 2020, the total FBD case count was the lowest in 23 years of monitoring at the national level. Suggesting that these changes are due to changing behavior in seeking health care international travel restrictions, among others [28]. The same assumption of the influence of the COVID-19 pandemic occurred in another study in the USA [29], and the studies in the United States and Canada emphasize that more studies are needed to confirm this phenomenon [29].

It is important to emphasize that the right to health is one of the social rights provided for in Brazil’s Federal Constitution (in articles 6 and 196) which imposes government and economic policies that are designed to promote, protect, and recover health, reducing the risk of disease and other health problems [30]. It is people’s right to expect that the food they eat is safe and suitable for them. Diseases and harm caused by food are, at best, unpleasant and, at worst, fatal [31].

Even though Brazilian legislation is one of the strictest, and despite the sanitary control measures that already exist in the country, Brazil continues to face problems in controlling FBDs throughout its territory, as evidenced by the data presented. The continental size of the country prevents inspection actions in all food-producing establishments and homes, as well as a lack of training and awareness among the staff responsible for filling in surveillance forms. However, it is suspected that the COVID-19 pandemic may have prompted greater caution and precision in the data on FBDs, as has been widely discussed. 

The limitation of this study is the short timeframe of only four years of data analyzed, as well as the reliability of the records of the records of cases of the pandemic. FBDs are diseases that are underreported or even incorrectly recorded by the health service, which is why longer periods of evaluation may be interesting. However, the aim of this study was to assess the burden resulting from the COVID-19 pandemic.

## 5. Conclusions

Given the global health mobilization proposed by the WHO to increase adherence to hand hygiene to save lives during the COVID-19 pandemic, it was believed that there would be a decrease in the incidence of FBDs in Brazil during the pandemic. However, it was found that the number of notifications remained the same, with a migration of locations, an increase in the number of cases in hospitals and health units, and a reduction in cases at social events because they were banned. 

The increase in notifications in hospitals and health units demonstrates the need to improve the knowledge, attitudes, and food safety practices of food handlers and health professionals as they deal with vulnerable patients with potential health risks.

We emphasize the importance of new studies that evaluate and compare the same data after the COVID-19 pandemic, as health systems return to pre-pandemic conditions, to provide evidence for the adequacy of public policies and future interventions.

## Figures and Tables

**Figure 1 nutrients-16-00060-f001:**
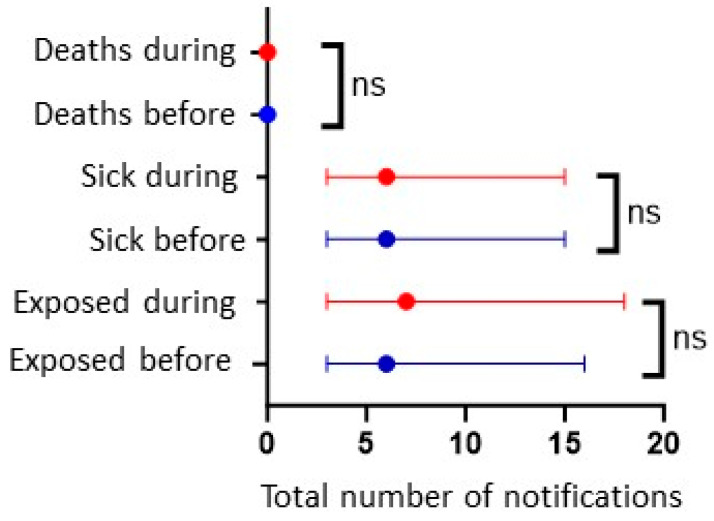
Median number of deaths, sicknesses, and exposures before and during the COVID-19 pandemic in Brazil. Label: ns *p* > 0.05.

**Figure 2 nutrients-16-00060-f002:**
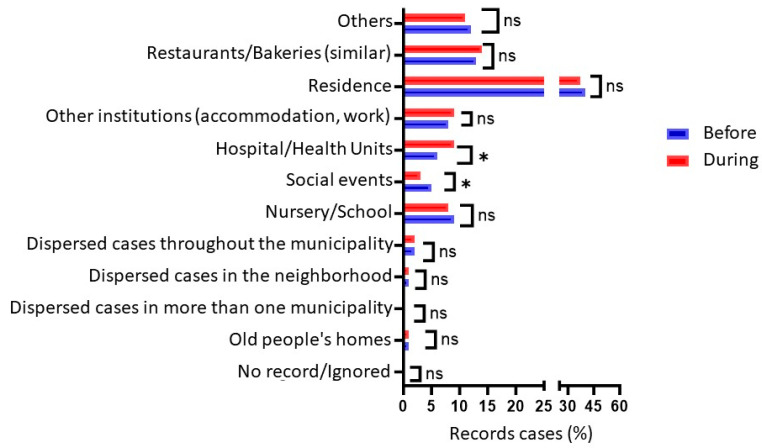
Pearson’s Chi-square of the locations of FBD outbreaks before and during the COVID-19 pandemic in Brazil. Label: * *p* < 0.05; ns *p* > 0.05.

**Figure 3 nutrients-16-00060-f003:**
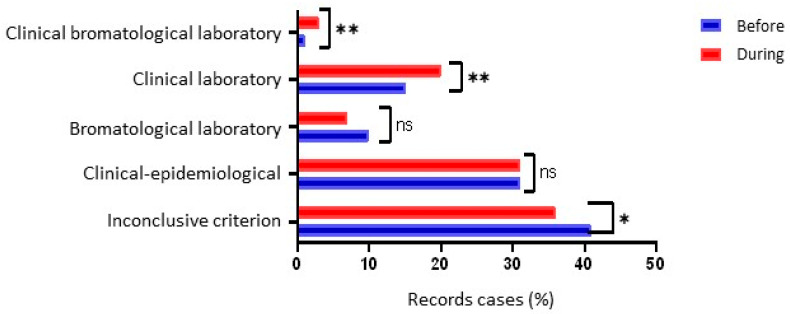
Pearson’s Chi-square between the types of tests used in FBD outbreaks before and during the COVID-19 pandemic in Brazil. Label: * *p* < 0.05; ** *p* < 0.001; ns *p* > 0.05.

**Table 1 nutrients-16-00060-t001:** Median, minimum, and maximum deaths, sickness, and exposures before and during the COVID-19 pandemic in Brazil.

	Time							
	Before the COVID-19 Pandemic				During the COVID-19 Pandemic			
	Median	Minimum	Maximum	Total Records at Time	Median	Minimum	Maximum	Total Records at Time
Total of exposures	6	0	36,566	1368	7	0	2900	838
Total of sicknesses	6	1	725	1368	6	1	404	838
Total of deaths	0	0	7	1368	0	0	4	838

**Table 2 nutrients-16-00060-t002:** Median, minimum, and maximum of deaths, sicknesses, and exposures per year before and during the COVID-19 pandemic in Brazil.

		Time							
		Before the COVID-19 Pandemic				During the COVID-19 Pandemic			
		Median	Minimum	Maximum	Total Records at Time	Median	Minimum	Maximum	Total Records at Time
Total deaths	2018	0	0	7	597				
	2019	0	0	2	771				
	2020					0	0	2	292
	2021					0	0	4	546
Total sicknesses	2018	6	2	725	597				
	2019	7	1	448	771				
	2020					5	2	350	292
	2021					6	1	404	546
Total exposures	2018	6	36,566	0	597				0
	2019	5	0	1697	771				0
	2020					6	0	2689	292
	2021					7	0	2900	546

## Data Availability

Data is contained within the article.

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
