# Peer review of "Foodborne Diseases: A Study before and during the COVID-19 Pandemic in Brazil"

_nutrients, 2023, doi:10.3390/nu16010060_

Round 1

Reviewer 1 Report

Comments and Suggestions for Authors

The authors decribe the reporting of foodborne diseasesin Brazil in the two years before the COVID-19 pandemic and two years of the pandemic. Overall, an interesting analysis has been done. Nevertheless, I have some questions and comments.

Major comments

1.       In the first sentence of the Introduction (lines 30-31) -and alo on several other places- the suggestion arises that SARS-CoV-2 is a foodborne disease. Most measures taken during the pandemic are also good for the prevention of foodborne diseases, but that is not the same. Please be more specific on this topic.

2.       On several places (e.g. lines 40-41, 46-50), the authors use a general “virus” where they mean SARS-CoV-2. Please be more specific, as for example in line 40-41 it now looks like transmission of virusses in the food chain does not happen, where this only applies to SARS-CoV-2.

3.       Results: please start the Results with overall numbers and figures from the database used, as (partly) is done in lines 162-178. Thus, please move these sentences to the beginning and consider to add a table with the main numbers as number of illnesses, deaths, etc per year.

4.       Discussion: the discussion is not coherent and therefore hard to follow. General points related to the results and effects of the pandemic are intertwined, as well as points that are made without any connection to the results or other points in the discussion.

a.       Please, consider rearranging the discussion interpreting the results with attention to overall points to make, differences between before and during pandemic, and effects of the pandemic.

b.       Sentences of which the meaning in relation to the results were not clear to me: lines 243-250 and 280-293.

Minor comments

5.       The abbreviation in line 75 “FDB” should be “FBD”? And please explain abbreviation in line 83 (ATD).

6.       Please add the reference directly after Draeger et al. (line 110)

7.       Line 126: Confidence intervals are normally 95% intervals. Nevertheless, as I don’t see any intervals given in the paper, I assume the authors mean that a treshold of significance (p-value) of 5% was used?

8.       Figure 3. The x-axis is not clear to me. Median number of sickness is 5, with a total number of records of 20? Is this per 100,000 inhabitants and/or per year, month, two-years period?

9.       Lines 263-265: please add the figure(s) of the period 2007-2017.

10.   Line 267: what is 30.1% and 8.8%? In the previous part of the paragraph the numbers per 100,000 inhabitants were given, what are these percentages?

11.   Lines 273-279. I don’t understand what the authors try to express here. Please consider rewriting these sentences.

Comments on the Quality of English Language

The structure of the sentences are not always correct, as there are several grammatical errors. I would suggest editing of the English language.

Author Response

Comments 1: In the first sentence of the Introduction (lines 30-31) - and all on several other places - the suggestion arises that SARS-CoV-2 is a foodborne disease. Most measures taken during the pandemic are also good for the prevention of foodborne diseases, but that is not the same. Please be more specific on this topic.

Response 1: Agree. We have, accordingly, changed the text (lines 31 - 40) to emphasize this point.

“COVID-19 is caused by a new coronavirus pathogen. The infection causes an acute febrile respiratory illness. People with COVID-19 can be asymptomatic and silently spread the disease (1). Contact with the contamination in the air environment is the main factor in the spread of the disease. Among the various potential atypical modes of transmission, food-borne transmission is widely discussed (2). Presently, there is no confirmation that COVID-19 is a food-borne disease. However, food contamination is a serious health and management problem. Since hand contamination can result in the transport of pathogens to the oral or nasal cavity, it has been widely reported that good hand hygiene during the COVID-19 outbreak, or not, was and is essential to prevent cross-contamination. An infected person can contaminate the entire surrounding environment (3)”.

Comments 2: On several places (e.g. lines 40-41, 46-50), the authors use a general “virus” where they mean SARS-CoV-2. Please be more specific, as for example in line 40-41 it now looks like transmission of virusses in the food chain does not happen, where this only applies to SARS-CoV-2.

Response 2: Agree. We have, accordingly, modified the text (lines 40 – 54) to emphasize this point.

This part has been updated to (lines 46-60):

“According to the Food and Agriculture Organization (FAO), there is no evidence of the COVID-19 virus transmission through the food chain (5). It is necessary to apply the principles of environmental sanitation, personal hygiene, and food safety practices to achieve prevention. The integrity of the food process depends on strict observance of good hygiene practices, environmental sanitation, and proper procedures for handling food, especially of animal origin (5).

In Brazil, the National Health Surveillance Agency (ANVISA) expanded its guidelines for food companies through Technical Notes (NT) No. 47, 48, and 49/2020, emphasizing that although there is no proof of transmission of the SARS-CoV-2 virus through food products, there was a need for greater attention to good manufacturing and product handling practices, with a focus on workers and handling environments(6,7). The recommendations call for the use of masks and gloves in food services, as well as an assessment of workers' health, personal hygiene, use of physical barriers, personal protective equipment, physical distancing, control of raw materials, controlled production flow, division of shifts for employees and transportation of products (6,7).”

Comments 3:   Results: please start the Results with overall numbers and figures from the database used, as (partly) is done in lines 162-178. Thus, please move these sentences to the beginning and consider to add a table with the main numbers as number of illnesses, deaths, etc per year.

Response 3: Thanks for the suggestion. We added 2 tables and referred to them within the text (lines 162-172).

“The medians, minimums and maximums of total deaths, illnesses and exposure before and during the COVID 19 pandemic can be seen in Table 1.”

“The medians, minimums and maximums, per year, of deaths, illness and exposure before and during the COVID 19 pandemic were also calculated, which can be seen in Table 2.”

Table 1. Median, minimum, and maximum deaths, sick and exposure before and during the COVID-19 pandemic, Brazil.

Time

Before the COVID-19 pandemic

During the COVID-19 pandemic

Median

Minimum

Maximum

Total records at time

Median

Minimum

Maximum

Total records at time

Total of deaths

0

0

 7

1368

0

0

2900

838

Total of sicks

6

1

725

1368

6

1

404

838

Total of exposed

6

0

36566

1368

7

0

4

838

Table 2 Median, minimum and maximum of deaths, sick and exposed deaths per year before and during the COVID – 19 pandemic, Brazil

Time

Before the COVID-19 pandemic

During the COVID-19 pandemic

Median

Minimum

Maximum

Total records at time

Median

Minimum

Maximum

Total records at time

Total of deaths

2018

0

0

7

597

2019

0

0

2

771

2020

0

0

2

292

2021

0

0

4

546

Total of sicks

2018

6

2

725

597

2019

7

1

448

771

2020

5

2

350

292

2021

6

1

404

546

Total of exposed

2018

6

0

36566

597

0

2019

5

0

1697

771

0

2020

6

0

2689

292

2021

7

0

2900

546

Comments 4: Discussion: the discussion is not coherent and therefore hard to follow. General points related to the results and effects of the pandemic are intertwined, as well as points that are made without any connection to the results or other points in the discussion.

  1. Please, consider rearranging the discussion interpreting the results with attention to overall points to make, differences between before and during pandemic, and effects of the pandemic.
  2. Sentences of which the meaning in relation to the results were not clear to me: lines 243-250 and 280-293.

Response 4: I agree. Thank you very much for your note, which has been promptly addressed. Therefore, the order in which the discussion was written has been redone, as well as lines 243-250 and 280-293.

has been updated to (lines 286-289):

“Most of the studies presented during this discussion always come to the same conclusion. They always agree that investments are needed in infrastructure (sanitation, water, etc.), in health surveillance of restaurants, hospitals and health services, and in educating the population (18-20)”.

has been updated to (lines 307-312):

“It is important to emphasize that the right to health is one of the social rights provided for in Brazil's Federal Constitution (in articles 6 and 196 which imposes on the government and economic policies that are designed to promote, protect, and recover health, reducing the risk of disease and other health problems (27). It is people's right to expect that the food they eat is safe and suitable for them. Diseases and harm caused by food are, at best, unpleasant and, at worst, fatal (28)”.

Comments 5: The abbreviation in line 75 “FDB” should be “FBD”? And please explain abbreviation in line 83 (ATD).

Response 5: Agree, thank you for pointing this out. We have, accordingly, revised and changed the abbreviation FDB and ATD, as they should be FBD. It was a typing mistake.

Comments 6: Please add the reference directly after Draeger et al. (line 110)

Response 6: Agree. We have, accordingly, done de reference directly after Draeger et al. as was requested (line 118).

Comments 7:  Line 126: Confidence intervals are normally 95% intervals. Nevertheless, as I don’t see any intervals given in the paper, I assume the authors mean that a tresh old of significance (p-value) of 5% was used?

Response 7: Sorry for our mistake. We added to the method the 95% confidence interval and the significance of the tests 5%.

Comments 8: Figure 3. The x-axis is not clear to me. Median number of sickness is 5, with a total number of records of 20? Is this per 100,000 inhabitants and/or per year, month, two-years period?

Response 8: Thanks for highlighting. The x-axis represents the number of notifications in the period.

Comments 9: Lines 263-265: please add the figure(s) of the period 2007-2017.

Response 9: Thanks for pointing that out. However, if we want to use a figure from another author's work, we will need this permission. Unfortunately, we don't have that authorization.

Comments 10: Line 267: what is 30.1% and 8.8%? In the previous part of the paragraph the numbers per 100,000 inhabitants were given, what are these percentages?

Response 10: Thank you for pointing this out. We agree with this comment. We also seek out this suggested information when writing the discussion, but the report doesn't provide it. It just says that: "The number of deaths decreased both in compared with 2020 (three fewer deaths; a relative reduction of 8.8%) and with the pre-pandemic years (13 fewer deaths; a relative reduction of 30.1% compared to 2017-2019)."

Comments 11:  Lines 273-279. I don’t understand what the authors try to express here. Please consider rewriting these sentences.

Response 11: Agree. We have, accordingly, modified the text (lines 233-239) to make the information more understandable.

“In the two periods evaluated in this study, the lethality rate remained at 0.5% per 100,000 inhabitants, demonstrating that the advent of the COVID-19 pandemic had no impact on the risk of death from FBD. The rate found in the study by Draeger et al. (2018) is lower, at 0.09% per 100,000 inhabitants over the 11 years investigated. This can probably be explained by the overload of health services with COVID-19, which reached more than 37 million cases, more than 700,000 deaths and a lethality rate of 1.9. These data shed light on what the priority was during the period in question (24). “

Reviewer 2 Report

Comments and Suggestions for Authors

Introduction

The research gap is not mentioned, so it would be good to have it. Have there been any previous studies of foodborne disease in the COVID-19 epidemic?

Method
Line 110 The authors mentioned they refer the protocol developed by Draeger.

However, citation could not be found in the manuscript.

What type of protocol is it?

Providing details of protocol would be helpful to understand the research methods.

Results
Figure 3 shows the death, sickness, and exposure.

What are the differences between sick and exposed?

How did the author define these?

Author Response

Comments 1: Introduction

The research gap is not mentioned, so it would be good to have it. Have there been any previous studies of foodborne disease in the COVID-19 epidemic?

Response: Thank you for pointing this out. We agree with this comment, accordingly, modified the text (lines 87-93) to emphasize this point.

“Until now, with a Brazilian national base, this is the first study to analyze data on FBD notifications, comparing the pre-pandemic periods (2018 and 2019) and during the COVID-19 health emergency (2020 and 2021), considering the implications of the COVID-19 pandemic scenario for food safety in Brazil, the consequent reorganization of health services, food production and consumption. Some other countries analyzed the pre-pandemic scenario and the scenario during the pandemic, but they did not analyze the period of two years before and two years during the pandemic”.

Comments 2: Method
Line 110 The authors mentioned they refer the protocol developed by Draeger.

However, citation could not be found in the manuscript.

What type of protocol is it?

Providing details of protocol would be helpful to understand the research methods.

Response 2: Agree. We have, accordingly, done de reference directly after Draeger et al. to helpful and understand the research methods (line 118).

Comments 3:   Results

Figure 3 shows the death, sickness, and exposure.

What are the differences between sick and exposed?

How did the author define these?

Response 3: Thank you. This is a very important point. We believe that explaining it will be an important methodological topic for the study and for readers. So, the difference between sick and exposed is included in the methodology part of the article (lines 126-144).

“Additionally, the Integrated Manual for the Prevention and Control of Foodborne Diseases is a document used to regulate the actions and instruments used in the investigation of FBD outbreaks, as well as guiding the information flow of the VE-TDA System, serving as a technical basis for the development of activities. This document defines those exposed as the group of people who participate in a meal which has caused an FBD outbreak (page 74), and sick people as the individuals who have symptoms and are related to the outbreak (pages 74 and 84). The report also informs that it is usual for an outbreak to have people who consume the food and do not become illness, which may be caused by the resistance and susceptibility of the host, or by consuming portions with non-infectious doses, among others”.

Round 2

Reviewer 1 Report

Comments and Suggestions for Authors

The authors have improved the manuscript significantly.

Comments on the Quality of English Language

The manuscript still needs editing of English language

Reviewer 2 Report

Comments and Suggestions for Authors

The manuscript seems to be revised adequately.